# Genome Insights into the Novel Species *Jejubacter calystegiae*, a Plant Growth-Promoting Bacterium in Saline Conditions

**Ling Min Jiang, Yong Jae Lee, Ho Le Han, Myoung Hui Lee, Jae Cheol Jeong, Cha Young Kim** [ID] **, Suk Weon Kim** [ID] **and Ji Young Lee \*** [ID]

Korean Collection for Type Cultures, Biological Resource Center, Korea Research Institute of Bioscience and Biotechnology (KRIBB), Jeongeup 56212, Korea; jiang6@kribb.re.kr (L.M.J.); tmx@kribb.re.kr (Y.J.L.); hlhan@kribb.re.kr (H.L.H.); mhlee17@kribb.re.kr (M.H.L.); jcjeong@kribb.re.kr (J.C.J.); kimcy@kribb.re.kr (C.Y.K.); kimsw@kribb.re.kr (S.W.K.)
\* Correspondence: jiyoung1@kribb.re.kr; Tel.: +82-63-570-5651

**Abstract:** *Jejubacter calystegiae* KSNA2[T], a moderately halophilic, endophytic bacterium isolated from beach morning glory (*Calystegia soldanella*), was determined to be a novel species in a new genus in the family *Enterobacteriaceae*. To gain insights into the genetic basis of the salinity stress response of strain KSNA2[T], we sequenced its genome using two complementary sequencing platforms (Illumina HiSeq and PacBio RSII). The genome contains a repertoire of metabolic pathways, such as those for nitrogen, phosphorus, and some amino acid metabolism pathways. Functional annotation of the KSNA2[T] genome revealed several genes involved in salt tolerance pathways, such as those encoding sodium transporters, potassium transporters, and osmoprotectant enzymes. Plant growth-promoting bacteria-based experiments indicated that strain KSNA2[T] promotes the germination of vegetable seeds in saline conditions. Overall, the genetic and biological analyses of strain KSNA2[T] provide valuable insights into bacteria-mediated salt tolerance in agriculture.

**Keywords:** genome sequence; *J. calystegiae*; salinity stress; secondary metabolites; plant growth promotion

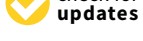



## 1. Introduction

Soil salinity can be a devastating environmental stress and cause major reductions in crop yields [1]. Salinity stress causes changes in various physiological and metabolic processes of the plant, inhibiting crop yield and quality [2]. Some microbes that inhabit halophytic plants may co-evolve with their host and enhance their tolerance to environmental stress [3–6]. These microbes can alleviate salinity stress by mitigating stress-induced physiological changes in plants through various mechanisms; for example, by synthesis of different phytohormones such as indole-3-acetic acid (IAA), 1-aminocyclopropane-1-carboxylate (ACC) deaminase, exopolysaccharides, volatile organic compounds, fixation of atmospheric nitrogen, and solubilization of mineral phosphate (P) [7]. Furthermore, salinity stress significantly affects carbohydrate, energy, and amino acid metabolism, and fatty acid biosynthesis. Therefore, it has been suggested that harnessing the potential of microbes is a promising approach to enhance plant growth in saline soil.

*Jejubacter calystegiae* KSNA2[T], a Gram-negative, facultatively anaerobic, motile, rod-shaped bacterium, was isolated from the stem endosphere of beach morning glory (*Calystegia soldanella*) on Chuja Island, Jeju-do, Korea. Strain KSNA2[T] was recently described as a novel species in a new genus in the family *Enterobacteriaceae* [8]. The strain tolerates sodium chloride concentrations up to 11% *w/v*. Some *Enterobacteriaceae* confer beneficial effects such as improving plant growth by inhibiting plant pathogens or fixing nitrogen [9,10], but the potential agricultural application of the new isolate has not been defined. Genome analyses provide insights into the physiological features and potential ecological roles of bacteria. To investigate the genetic basis and molecular mechanism of

how strain KSNA2[T] might interact with plants, we sequenced and annotated the strain's genome. The complete genome sequence of *J. calystegiae* KSNA2[T] provides a framework for further genetic studies to better understand the mechanism of bacteria-mediated salt tolerance in plants.

## 2. Materials and Methods

### 2.1. Bacterial Strain, Growth Conditions, and Genomic DNA Extraction

Strain KSNA2[T] was maintained routinely on Tryptic Soy Agar (TSA, Difco Laboratories, Detroit, MI, USA) at 25 °C for 2 days, and stored longer term in sterile glycerol (15%, *w/v*) at −80 °C. A single colony from TSA was inoculated into 1 L of Tryptic Soy Broth (TSB, Difco) and incubated at 25 °C for 3 days with shaking at 150 rpm. Cells were collected by centrifugation at 8000 rpm, and genomic DNA was extracted using a genomic DNA purification kit (MGmed, Seoul, Korea) following the manufacturer's protocol.

### 2.2. Whole-Genome Sequencing and Assembly

The genome of KSNA2[T] was sequenced as previously described [8]. A total of 1,384,706,260 sub-read bases (mean sub-read length 9692) and 142,866 sub-reads ($N_{50}$ value of 13,531) were generated. Error correction was performed using Pilon (v1.21) to obtain a high-quality sequence. After HiSeq raw data filtering, 920,878,332 total read bases, 6,098,532 total reads, and a 94.9% ratio of bases with a Phred quality score over 30 were generated.

### 2.3. Genome Annotation and Property Analysis

Gene prediction was accomplished using the Rapid Annotation using Subsystem Technology SEED viewer (RAST; http://rast.nmpdr.org/) [11]. Metabolic pathway analysis was carried out using the KEGG (Kyoto Encyclopedia of Genes and Genomes) GENOME database [12]. AntiSMASH (v.5.1.1) was used to identify secondary metabolite biosynthesis gene clusters [13]. In addition, gene functions were predicted through the cluster of orthologous groups (COG) database [14], and the genome sequence was screened for virulence and antibiotic resistance using VirulenceFinder and ResFinder (https://cge.cbs.dtu.dk/services/) [15].

### 2.4. Evaluation of Strain KSNA2 on Seed Germination in Saline Conditions

Seeds of cabbage, pepper, and carrot were used in this study. The seeds were surface sterilized with 1% (*v/v*) sodium hypochlorite in water for 10 min followed by rinsing 5 times in distilled water. The seeds were then soaked in a KSNA2[T] suspension ($10^8$ CFU/mL) or TSB (control) for 8 h. A total of 20 seeds were sown in Petri dishes (90 mm diameter) with two sheets of sterile filter paper (Advantech, Tokyo, Japan). The seeds were moistened either with distilled water (control) or with solutions of varying NaCl concentrations (50, 100, 200, and 300 mM) and kept at 25 °C. After 7 d, the emergence of the radicle from the seeds was considered indicative of germination. The germination percentage was calculated as follows: germination % = (number of seeds germinated/number of seeds sown) × 100. All experiments were performed in triplicate.

### 2.5. Nucleotide Sequence Accession Numbers

The complete genome sequence of *J. calystegiae* KSNA2[T] has been deposited in GenBank under accession number CP040428. The strain is available at the Korean Collection for Type Cultures (accession number KCTC 72234[T]) and at the China Center for Type Culture Collection (accession number CCTCC AB 2019098[T]).

## 3. Results and Discussion

### 3.1. Genomic Features

To investigate the potential functional capabilities of strain KSNA2[T], whole-genome sequencing was performed using a combination of PacBio RSII and Illumina HiSeq X-ten

sequencing platforms [8]. PacBio RSII produces long reads to facilitate de novo assembly of long repetitive sequences, but the required coverage is more costly to attain; conversely, the Illumina HiSeq X-ten produces short-read sequences, though the read length can limit assembly. However, this platform can handle lower quantity and quality genomic DNA than the PacBio RSII [16]. In our study, we combined both platforms to get high quantity and quality genome sequences. As described previously [8], the genome of strain KSNA2^T comprises a single circular chromosome of 5,182,800 bp with a G + C content of 56.1%. The genome was predicted to contain 4679 protein-coding genes, 83 transfer RNA genes, 22 ribosomal RNA genes, 7 ncRNA genes, and 97 pseudogenes. A circular genome map was generated using the GC view program [17] (Figure 1).

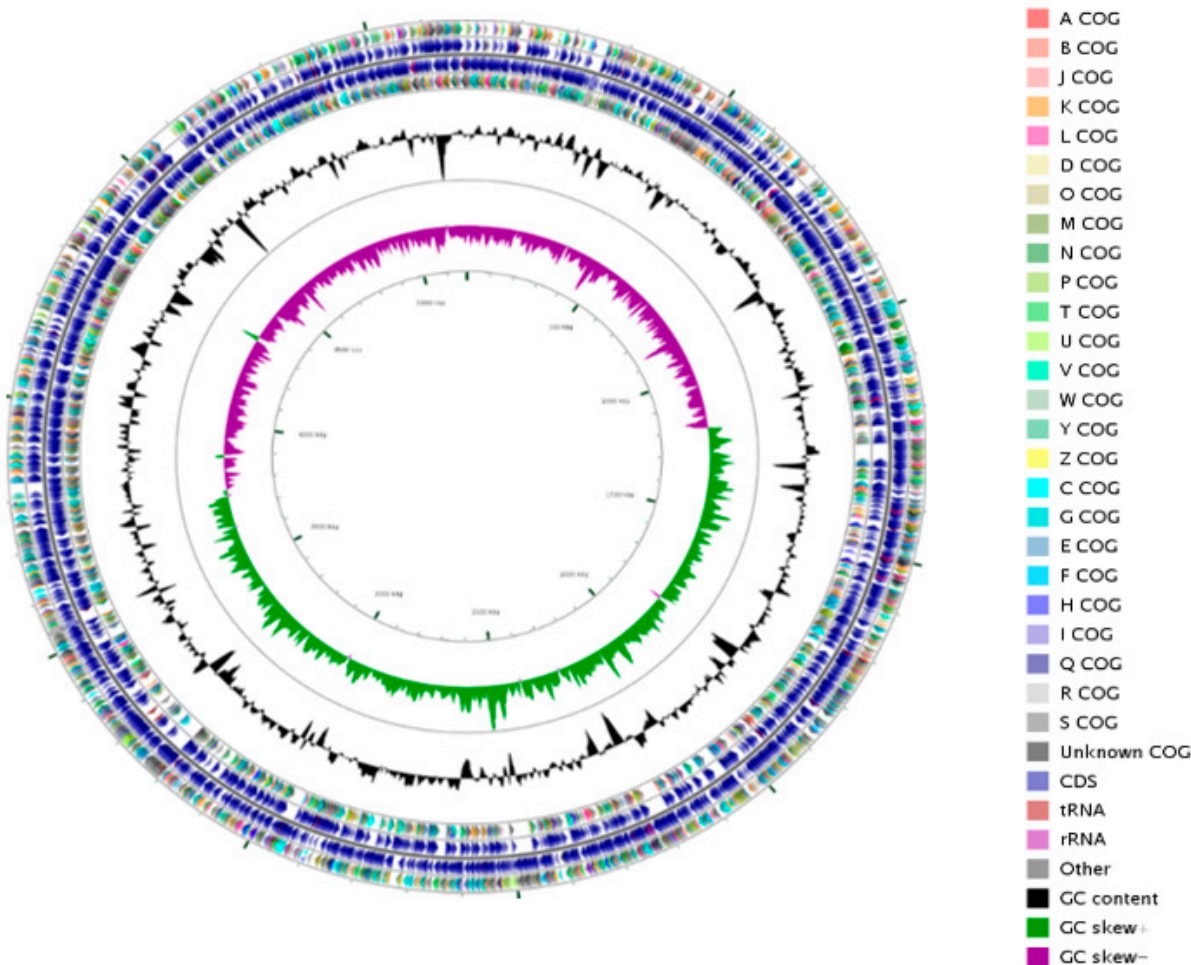

**Figure 1.** Map of the *J. calystegiae* KSNA2^T genome. Marked characteristics are shown from the outside to the center. Rings 1 and 4 show COG annotation in the forward and reverse directions, respectively. Rings 2 and 3 show the PGAP annotation performed by NCBI in the forward and reverse directions, respectively. Ring 5 shows the G + C% content plot. The innermost ring shows the GC skew. The COG categories are: A, RNA processing and modification; B, chromatin structure and dynamics; C, energy production and conversion; D, cell cycle control, cell division, chromosome partitioning; E, amino acid transport and metabolism; F, nucleotide transport and metabolism; G, carbohydrate transport and metabolism; H, coenzyme transport and metabolism; I, lipid transport and metabolism; J, translation, ribosomal structure, and biogenesis; K, transcription; L, replication, recombination, and repair; M, cell wall/membrane/envelope biogenesis; N, cell motility; O, posttranslational modification, protein turnover, chaperones; P, inorganic ion transport and metabolism; Q, secondary metabolite biosynthesis, transport, and catabolism; R, general function prediction only; S, function unknown; T, signal transduction mechanisms; U, intracellular trafficking, secretion, and vesicular transport; V, defense mechanisms; W, extracellular structures; X, mobilome: prophages, transposons; and Z, cytoskeleton.

### 3.2. Genes of Central Metabolism and Cellular Processes

Cluster orthologous group (COG) analysis revealed that the largest group of coding sequences (CDSs) was classified as function unknown (28.1% of the total assigned COGs), followed by those involved in amino acid transport and metabolism (8.1%), and carbohydrate transport and metabolism (7.7%) (Figure 1 and Table 1). Carbon and nitrogen sources are essential nutrients for bacterial growth as well as protein biosynthesis in all organisms. Additionally, nitrogen regulatory protein genes (*ptsN*, *glnK*, *glrR*), urease genes (*ure* ABCDEFG), and carbonic anhydrase (*can*, *cah*) were present, indicating that strain KSNA2^T might play an important role in the nitrogen and carbon cycles. Analysis of the whole genome using VirulenceFinder and ResFinder revealed no known virulence and antibiotic resistance genes.

**Table 1.** COG functional categories of the genome sequence of *J. calystegiae* KSNA2^T.

| | Categories | Count | Ratio (%) |
|---|---|---|---|
| | Information storage and processing | | |
| J | Translation, ribosomal structure, and biogenesis | 184 | 3.9561 |
| A | RNA processing and modification | 1 | 0.0215 |
| K | Transcription | 344 | 7.3963 |
| L | Replication, recombination, and repair | 185 | 3.9776 |
| | Cellular process and signaling | | |
| B | Chromatin structure and dynamics | 0 | 0.0000 |
| D | Cell cycle control, cell division, chromosome partitioning | 41 | 0.8815 |
| Y | Nuclear structure | 0 | 0.0000 |
| V | Defense mechanisms | 58 | 1.2470 |
| T | Signal transduction mechanisms | 149 | 3.2036 |
| M | Cell wall/membrane/envelope biogenesis | 265 | 5.6977 |
| N | Cell motility | 48 | 1.0320 |
| Z | Cytoskeleton | 0 | 0.0000 |
| W | Extracellular structures | 0 | 0.0000 |
| U | Intracellular trafficking, secretion, and vesicular transport | 80 | 1.7201 |
| O | Posttranslational modification, protein turnover, chaperones | 140 | 3.0101 |
| | Metabolism | | |
| C | Energy production and conversion | 291 | 6.2567 |
| G | Carbohydrate transport and metabolism | 357 | 7.6758 |
| E | Amino acid transport and metabolism | 379 | 8.1488 |
| F | Nucleotide transport and metabolism | 100 | 2.1501 |
| H | Coenzyme transport and metabolism | 142 | 3.0531 |
| I | Lipid transport and metabolism | 102 | 2.1931 |
| P | Inorganic ion transport and metabolism | 266 | 5.7192 |
| Q | Secondary metabolites biosynthesis, transport, and catabolism | 75 | 1.6126 |
| | Poorly characterized | | |
| R | General function prediction only | 139 | 2.9886 |
| S | Function unknown | 1305 | 28.0585 |
| Totals | | 4651 | 100 |

### 3.3. Genes Associated with Multi-Stress Response

The Rapid Annotation using Subsystem Technology SEED annotation (RAST; http://rast.nmpdr.org/) revealed that 3222 genes of CDSs were not in a subsystem (1815 non-hypothetical; 1407 hypothetical). RAST annotation revealed the involvement of 89 genes in stress responses, including 21 in osmotic stress, 30 in oxidative stress, 16 in detoxification, 24 in general stress response, and 7 in the periplasmic stress response (Figure S1 and Table S1).

### 3.4. Secondary Metabolites and Salt Tolerance Pathway

There were 4122 protein-coding genes connected to the Kyoto Encyclopedia of Genes and Genomes (KEGG) pathways. These were analyzed using Blast KOALA based on

the KEGG pathway database and contained annotated genes in 123 metabolic pathways, including the TCA cycle, pentose-phosphate pathway, and galactose metabolism pathway. KSNA2$^T$ can utilize carbon sources such as pentose, fructose, galactose, and starch-sucrose. Metabolic genes, including those involved in the fatty acid, sulfur, nitrogen, biotin, vitamin B6, riboflavin, thiamin, and methane metabolic pathways, were present. In addition, 13 membrane transport and signal transduction pathways, including ABC transport, phosphotransferase system (PTS), bacterial secretion system, two-component system, mitogen-activated protein kinase (MAPK) signaling pathway-fly, MAPK signaling pathway-plant, MAPK signaling pathway–yeast, HIF-1 signaling pathway, FoxO signaling pathway, phosphatidylinositol signaling system, phospholipase D signaling pathway, PI3K-Akt signaling pathway, and 5′ adenosine monophosphate-activated protein kinase (AMPK) signaling pathway, were annotated. ABC-2 and other transporters, such as Na$^+$ transporters (NatB and NatA), were also found. Database analyses, a simplified model of metabolism, and the important cellular components of *J. calystegiae* KSNA2$^T$ are provided (Figure 2).

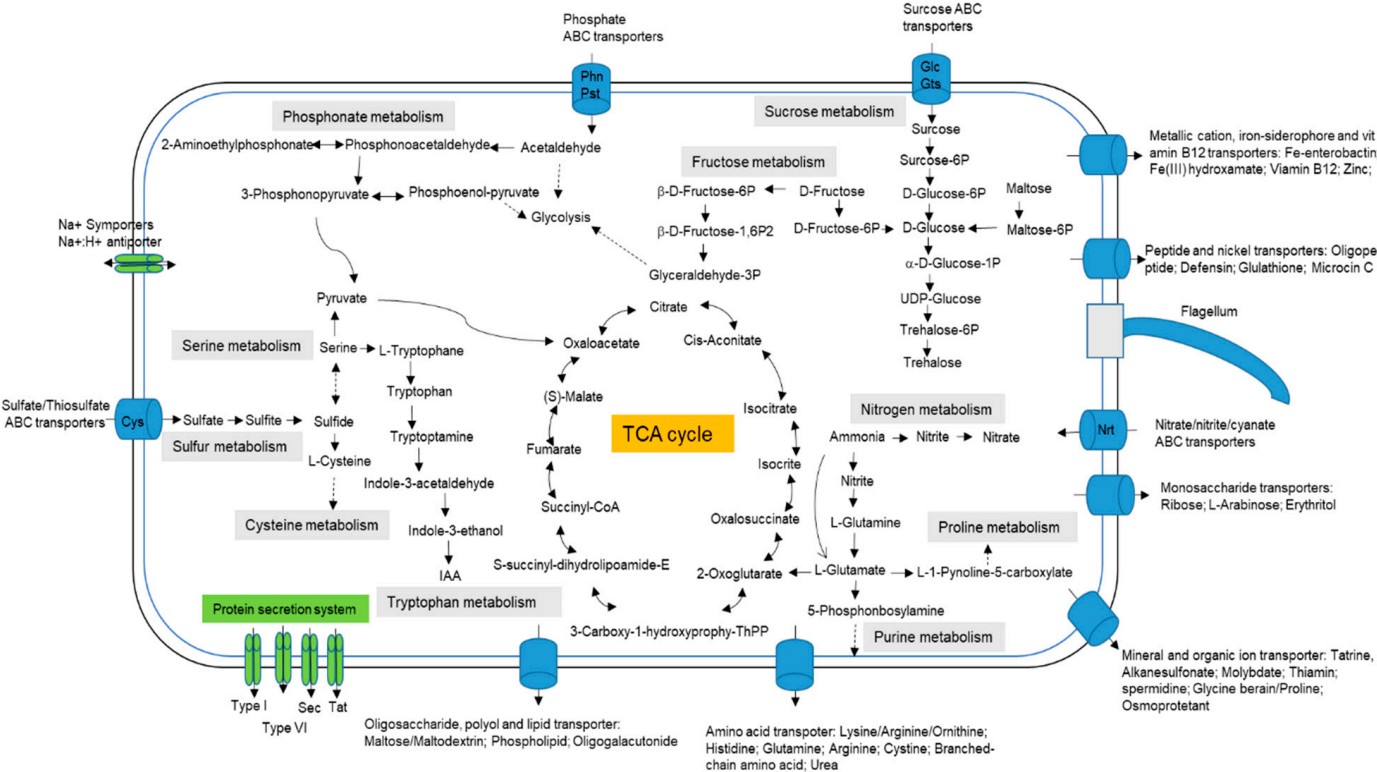

**Figure 2.** Metabolic pathways of strain KSNA2$^T$, including the TCA cycle, nitrogen metabolism, phosphate metabolism, sulfur metabolism, and sucrose metabolic pathway. The presence and absence of genes were predicted in Blast KOALA based on the KEGG pathway database.

Seven secondary metabolite regions were identified by AntiSMASH [13], including bacteriocin, acyl-amino-acids, ectoine, ladderance, butyrolactone, NRPS, thiopeptide, arylpolyene, and terpene. Among these, four clusters have high homology to known biosynthetic gene clusters (BGCs) encoding kosinostain, turnerbactin, O-antigen, and aryl polyenes (Table S2). The ectoine synthase (*ectC*) gene and ectoine hydrolase (*doeA*) gene were present in strain KSNA2$^T$, which could contribute to its osmotic and salt stress tolerance [8].

According to the KEGG Ontology annotation, the genes involved in the salt tolerance pathway are common in sodium transporters, including sodium ion transporters, ion exchangers, and sodium symporter. Some of the potassium transporters and osmoprotectant enzymes were also annotated in the genome (Table S3).

### 3.5. Strain KSNA2$^T$ Improves Seed Germination in Saline Conditions

To determine whether the strain KSNA2$^T$ affected the germination of vegetable seeds (cabbage, pepper, and carrot) in saline conditions, sterilized seeds were inoculated with a bacterial suspension. The seeds were then germinated on separate filter paper wetted with 10 mL of sodium chloride solutions of different concentrations (0, 50, 100, 200, and 300 mM), equivalent to 0%, 0.29%, 0.58%, 1.16%, and 1.75% (*w/v*) NaCl. The optimal NaCl tolerance for strain KSNA2$^T$ is 0–7%, so the strain's growth is not adversely affected below 2% (*w/v*) NaCl (Figure S2). The germination rates of cabbage and carrot seeds were greatly reduced from the 50 to 300 mM NaCl solution, but this inhibition was recovered by strain KSNA2$^T$ treatment in seeds grown from 100 to 200 mM NaCl (Figure 3a–c). Although the germination rate of pepper seeds did not show significant differences with or without strain KSNA2$^T$ treatment (75% and 80% in 100 mM NaCl, respectively) in saline conditions (Figure 3d), plant root growth of pepper by strain KSNA2$^T$ was faster than without the bacterial suspension in saline conditions (Figure S3).

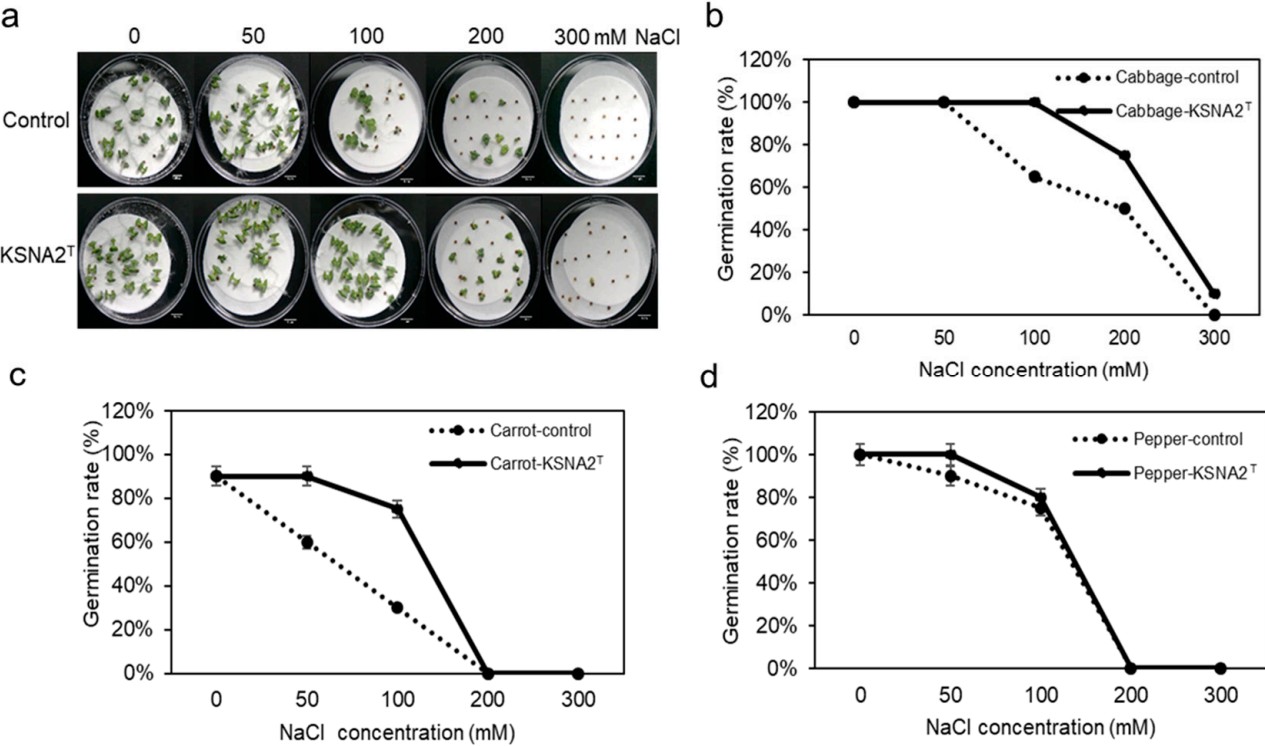

**Figure 3.** Effects of treatment with strain KSNA2$^T$ on seeds grown in different salt concentrations. The germination rates of seeds in various concentrations of NaCl (0, 50, 100, 200, and 300 mM) were calculated after seven days. Seeds were inoculated with the KSNA2$^T$ bacterial suspension ($10^8$ CFU/mL) and without (Tryptic Soy Broth (TSB) media, control) for 12 h. (**a**) Seedling growth in cabbage. The average germination rate of (**b**) cabbage, (**c**) carrot, and (**d**) pepper. Scale bar = 1 cm.

These results indicate that strain KSNA2$^T$ could improve seed germination in saline conditions, demonstrating its potential for future applications in agriculture and to provide valuable insights for studies of the mechanisms of salt tolerance in plants.

## 4. Discussion

### 4.1. Potential Ecological Role of Strain KSNA2$^T$

A previous study proved that plant-associated bacteria, especially endophytic bacterial populations, determine plant health status by increasing the plant utilization of nutrients such as nitrate uptake, phosphate solubilization, IAA production [18–21], and inhibition of phytopathogens [22]. Plants and microbes can co-evolve to enhance stress tolerance when

environmental conditions are harsh [3–5]. Here, strain KSNA2[T] was isolated from beach morning glory (*Calystegia soldanella*), a common pantropical creeping vine that is adapted to grow on dunes and even under extreme growth conditions such as scarce nutrients, salt spray, high temperature, excessive drought, blowing sand, and high winds [23,24]. Although bacterial diversity plays a vital role in the life of plants in the dune field, studies of the microbial communities associated with such plants are still incipient. This is especially so in terms of the mechanisms of the interactions between the microbes and the plants. According to colony pigmentation and morphology, we cultivated 126 different endophytic bacteria from the leaves, stems, and root parts of the beach morning glory, including strain KSNA2[T], the type strain of a novel species in a new genus; KSNA2[T] is able to grow between 4 and 45 °C, between pH 5.0 and 12.0, and in sodium chloride concentrations of 0 to 11% *w/v* (Figure S2 [8]).

Whole-genome analysis of strain KSNA2[T] showed that amino acid transport and metabolism, and carbohydrate transport and metabolism, dominate the major groups of proteins (Figure 1 and Table 1). These are required for growth and protein biosynthesis in all organisms. A large portion of genes encoding for proteins in the nitrogen metabolism pathway was represented throughout the whole genome (Table 1). The presence of genes that encode for proteins involved in nitrogen metabolism infer the potential for KSNA2[T] to be involved in various nitrogen cycles. Phosphate metabolic pathways in strain KSNA2[T] were also found throughout the genome, including the phosphate ABC transporters (phn and pst), and phosphonate metabolism (Figure 2). Phosphate-solubilizing bacteria (PSB) improve plant growth by dissolving insoluble phosphate. Overall, the whole genome information of strain KSNA2[T] confirmed that the strain plays crucial roles in plant growth-promoting roles such as nitrogen fixation and phosphate solubilization.

### 4.2. Potential Role of Strain KSNA2[T] as an Agricultural Reagent

Several reports have shown halotolerant PGPRs (plant growth-promoting rhizobacteria), which effectively improve the growth of various crops in saline conditions [21,25–29]. Single or consortium of halotolerant bacteria contributed to salinity tolerance of plants [30–33]. The salt tolerance mechanisms by halotolerant PGPRs have been shown to involve increasing the efficiency of inoculated plants to absorb ions to maintain a high $K^+/Na^+$ ratio. This can improve the nutritional status of both macronutrients and micronutrients by directly reducing the accumulation of toxic ions such as $Na^+$ and $Cl^-$ and regulating ion transporter expression and/or activity. It also reduces plant $Na^+$ accumulation by excreting exopolysaccharides (EPS), binding cations (especially $Na^+$) at the roots, and preventing translocation to the leaves [31–33]. The ACC deaminase-producing PGPRs reduce ethylene synthesis and enhance the availability of indole-3-acetic acid (IAA), zeatin (Zt), and gibberellin (GA) in plants in saline conditions. The halotolerant PGPRs also induce production of antioxidants for scavenging reactive oxygen species (ROS) generated in the plant under salinity stress [27–30].

Whole-genome analysis indicated that salt tolerance mechanism of strain KSNA2[T] may be attributed to osmotic balance, ion homeostasis: $Na^+/Ca^{2+}$ exchanger, sodium symporter, and potassium transporters, to pump $Na^+$ out of their membrane or balance the solute gradient by accumulating potassium. Multiple genes related to stress responses have been found to cooperate in conferring salt tolerance [34,35]. When bacteria suffer hyperosmotic stress, the flux of ions across their cellular membrane is controlled and osmolytes are accumulated to balance the solutes [36]. Genes encoding transporters and the synthesis of osmoprotectant enzymes are then triggered, porin expression is regulated by the EnvZ/OmpR two-component system, and salt ions (e.g., $K^+$, $Na^+$, $SO_4^{2-}$, $Cl^-$, and $Ca^{2+}$) or osmoprotectants such as choline and betaine, accumulate [37–39]. Genes encoding proteins involved in osmotic balance and ion homeostases, such as sodium transporters ($Na^+/Ca^{2+}$) exchanger, sodium symporter, $Na^+/H^+$ antiporters, and potassium transporters, and which can moderate the solute gradient by accumulating potassium, were also found in the annotated KSNA2[T] genome. Strain KSNA2[T] may also produce EPS which is important in

alleviating salinity stress due to EPS binding with $Na^+$ and decreasing the bioavailability of the ion for plant uptake [29,31]. Other metabolic pathways such as those involved nitrogen, tryptophan, sulfur, and phosphate metabolism were annotated in the KSNA2[T] genome (Figure 2); such pathways may also contribute to the promotion of plant growth.

These analyses may explain the capacity of a plant, beach morning glory, to grow in harsh environments. In crop germination tests under salt stress in vitro, germination rates were enhanced by treatment with strain KSNA2[T]. These data demonstrate that strain KSNA2[T] may be able to promote salt resistance in seeds and that it has potential for future applications in agriculture.

## 5. Conclusions

Whole-genome sequencing of the novel *J. calystegiae* KSNA2[T] revealed that it contained many genes encoding for proteins involved in nitrogen and phosphate metabolism, indicating that strain KSNA2[T] may play an important role in the nitrogen cycle and phosphate solubilization. A large number of interesting genes associated with multi-stress responses, including osmotic stress, oxidative stress, and 123 metabolic pathways that synthesize secondary metabolites, could yield potential applications in biotechnology. In particular, gene clusters encoding salt tolerance transporters can provide valuable insights into bacteria-mediated salt tolerance in agricultural applications. More research should be conducted to confirm the expression of salt-associated genes under salt stress. We conclude that the novel bacterium *J. calystegiae* KSNA2[T] has potential applications in agriculture.

**Supplementary Materials:** The following are available online at https://www.mdpi.com/1424-2818/13/1/24/s1, Figure S1: Distribution and count of the subsystem categories of *J. calysteage* KSNA2[T] based on RAST annotation, Figure S2: Growth of KSNA2[T] in 0 to 12.5% (*w/v*) NaCl, Figure S3: Effects of treatment with strain KSNA2[T] on pepper seeds grown in different salt concentrations (0, 50, 100, 200, and 300 mM NaCl), Table S1: Stress response genes annotated by RAST SEED server, Table S2: Predicted secondary metabolite loci in *J. calystegiae* KSNA2[T], and Table S3: Predicted stress response proteins in *J. calystegiae* KSNA2[T].

**Author Contributions:** Conceptualization, L.M.J. and J.Y.L.; methodology, Y.J.L. and H.L.H.; resources, M.H.L.; writing—original draft preparation, L.M.J. and J.Y.L.; writing—review and editing, J.C.J., C.Y.K. and S.W.K.; project administration, J.Y.L.; funding acquisition, C.Y.K. All authors have read and agreed to the published version of the manuscript.

**Funding:** This research was supported by the Basic Science Research Program through the National Research Foundation of Korea (NRF), funded by the Ministry of Education (RBM0142011), and by the KRIBB research initiative program.

**Conflicts of Interest:** The authors declare no conflict of interest.

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
