# Peer review of "Genome Insights into the Novel Species Jejubacter calystegiae, a Plant Growth-Promoting Bacterium in Saline Conditions"

_diversity, doi:10.3390/d13010024_

Round 1

Reviewer 1 Report

The manuscript “Genome Insights into the Novel Species Jejubacter calystegiae, a Plant growth-promoting Bacterium under Salinity Condition” deals with the complete genome sequence of the Jejubacter calystegiae using two sequencing platforms, and describes its involvment in the germination promotion of vegetable seeds under saline conditions.

The methodology and the data are well supported, anyway the manuscript should be revised before publication.

In my opinion there is only one point that I would like the authors to expand: a more deep comparison and differences between the to sequencing platform used, before to start with the genomic annotation and the following genes association to the different functional pathways.

Reviewer 2 Report

Review Genome Insights into the Novel Species Jejubacter calystegiae, a Plant growth-promoting Bacterium under Salinity Condition

The authors present data analysis of their previously published genome of Jejubacter calystegiae and seed germination data. The paper is well written, but some major issues need to be addressed before publication.

Since the authors published [Jiang et al., 2020. Jejubacter calystegiae gen. nov., sp. nov., moderately halophilic, a new member of the family Enterobacteriaceae, isolated from beach morning glory. Journal of Microbiology, 58(5), pp.357-366] and show in that paper whole genome sequencing and data analysis, this cannot be repeated in this manuscript. Lines 58-90 of the methods and 97-144 of the Results should be removed and referenced.

Figure 1 does not contribute to the paper. What does it tell the reader? Perhaps a comparative genomics figure using Mauve or Brig would benefit the paper.

Lines 195-199, the Antismash analysis should be expanded with more details, including similarity to other BGCs in which organisms.

Lines 220-240, this is interesting data. It would however be more convincing if the authors would include control experiments with a salt-tolerant but not PGPB, like some Vibrio species.

Line 224, the authors do not show any growth data of the bacteria at these salt concentrations. It would help the paper if the authors would include this.

Line 230, it would be beneficial to the paper to include the ‘data not shown’ for the pepper plant growth.

Line 252, do the authors mean strains or species here?

Mechanism (lines 272-290) should be discussed in more detail, taking into account the different mechanism that have been proposed for other PGPBs (e.g. see Kumar et al. 2020 and other recent reviews).

It would be advisable to review the recently published literature on PGPBs in the context of salinity protection and reference:

Kumar, A., Singh, S., Gaurav, A.K., Srivastava, S. and Verma, J.P., 2020. Plant Growth-Promoting Bacteria: Biological Tools for the Mitigation of Salinity Stress in Plants. Frontiers in Microbiology, 11.

Etesami, H. and Beattie, G.A., 2018. Mining halophytes for plant growth-promoting halotolerant bacteria to enhance the salinity tolerance of non-halophytic crops. Frontiers in microbiology, 9, p.148.

Numan, M., Bashir, S., Khan, Y., Mumtaz, R., Shinwari, Z.K., Khan, A.L., Khan, A. and Ahmed, A.H., 2018. Plant growth promoting bacteria as an alternative strategy for salt tolerance in plants: a review. Microbiological research, 209, pp.21-32.

And other papers published in 2019 and 2020.

Round 2

Reviewer 2 Report

The authors have satisfactory addressed all concerns and comments.